# Job Vacancy Ranking with Sentence Embeddings, Keywords, and Named Entities

Natalia Vanetik * and Genady Kogan

Department of Software Engineering, Shamoon College of Engineering, Beer Sheva 84500, Israel; genadko@ac.sce.ac.il
* Correspondence: natalyav@sce.ac.il; Tel.: +972-8-647-5015

**Abstract:** Resume matching is the process of comparing a candidate's curriculum vitae (CV) or resume with a job description or a set of employment requirements. The objective of this procedure is to assess the degree to which a candidate's skills, qualifications, experience, and other relevant attributes align with the demands of the position. Some employment courses guide applicants in identifying the key requirements within a job description and tailoring their experience to highlight these aspects. Conversely, human resources (HR) specialists are trained to extract critical information from numerous submitted resumes to identify the most suitable candidate for their organization. An automated system is typically employed to compare the text of resumes with job vacancies, providing a score or ranking to indicate the level of similarity between the two. However, this process can become time-consuming when dealing with a large number of applicants and lengthy vacancy descriptions. In this paper, we present a dataset consisting of resumes of software developers extracted from a public Telegram channel dedicated to Israeli hi-tech job applications. Additionally, we propose a natural language processing (NLP)-based approach that leverages neural sentence representations, keywords, and named entities to achieve state-of-the-art performance in resume matching. We evaluate our approach using both human and automatic annotations and demonstrate its superiority over the leading resume–vacancy matching algorithm.

**Keywords:** resume matching; keywords; named entities; semantic vectors

## 1. Introduction

The resume–vacancy matching problem refers to the task of automatically matching job seekers' resumes or CVs with job vacancies or job descriptions [1]. Its goal is to determine the degree of compatibility between a candidate's skills, qualifications, and experience and the requirements and preferences specified by the employer in the vacancy description. The automatization of a resume–vacancy ranking can help streamline the selection process, save time for both job seekers and employers, and improve the overall efficiency of the job market [2].

This task is not straightforward but has many different aspects or facets to consider because it is a complex process that requires careful attention to various factors. It involves comparing and matching different elements from the candidate's resume with the relevant parts of the job description. These elements usually include skills, experience, and keywords. Skills that the candidate possesses should align with the skills required for the job, and the candidate's past work experience must be assessed to see if it matches the vacancy experience requirements. In many cases, job descriptions include specific keywords that indicate essential qualifications. Resume–vacancy matching usually requires identifying these keywords in the resume and seeing if they match the ones listed in the job description. However, in most cases, a perfect match may not be possible. Some aspects of the candidate's qualifications might align with the job description, while others may not be an exact fit. Beyond simple matching, advanced techniques need to be applied to understand the semantics and context of the job requirements and the candidate's qualifications [3].

To address the resume–vacancy ranking problem, various automated techniques have been applied in recent years, including natural language processing, machine learning, and information retrieval. The key issues for these systems are that resumes come in a variety of formats, making it difficult to accurately extract relevant information. Furthermore, the job market moves quickly, and matching systems must react to changing job criteria and skill expectations.

Rule-based systems are relatively easy to develop and implement because they rely on predefined rules and keywords to match resumes with job vacancies. Recruiters can tailor these algorithms to specific job requirements, resulting in more accurate matches for their company. On the other hand, rule-based systems lack the agility to handle more nuanced matching circumstances and may struggle with unusual or fast-changing work needs. If applicants use different languages or synonyms for the same talents, over-reliance on keywords may result in false positives or mismatches. For example, Resumix [4] is one of the earliest systems that used rule-based methods to match candidate resumes with job vacancies based on predefined keywords and criteria. Works [5,6] use taxonomies and ontologies as the means of defining rules for resume–vacancy matching.

NLP-based systems leverage natural language processing techniques to understand the context and semantics of resumes and job descriptions, leading to more accurate matches. They can identify related skills and experiences even when specific keywords are not explicitly mentioned, making them more robust in matching candidates. However, biased training data can influence these systems, perhaps discriminating against specific groups or preferring applicants with certain attributes. The development of these systems necessitates substantial computer resources, and fine tuning the models can be time-consuming. For example, paper [7] represents text data with the help of word embeddings and the bag-of-words (BOW) model, while the authors of [8,9] used the BOW model with term frequency–inverse document frequency (TF-IDF) weights [10]. Paper [11] represents text with word n-gram vectors [12]. In general, all these methods use vector similarity for the chosen text representation to rank resumes or vacancies. Different similarity measures appear in these works, such as cosine similarity, Dice similarity [13], Jaccard similarity, and Jaccard overlap. Several advanced approaches use recurrent neural networks (RNNs) or convolutional neural networks (CNNs) to operate as feature extractors and a word embedding layer [14] to represent texts [15,16]. Our proposed method belongs to this category of resume-matching systems.

The next class of resume-matching systems is the general machine learning models. As more data are processed and input from recruiters is received, these models can continuously improve their matching accuracy. These systems can adjust to individual recruiters' or organizations' preferences, adapting matches to their own requirements. However, in order to work well, these systems require a vast amount of data, creating issues about data privacy and security. Some of the algorithms work as black boxes, making it difficult to comprehend how particular matches are formed, thus leading to biases. Examples of such systems include [17,18], which employ neural networks to create end-to-end matching models. Multiple works, such as [19–22], treat the task of resume matching as a recommender system that suggests the best vacancies for a given resume. The authors of [23] utilize historical data to facilitate the matching.

OKAPI BM25 [24] is a ranking algorithm used in information retrieval systems to rank documents based on their relevance to a given query. It is a good baseline for resume rating tests [25] and is one of the most accurate computer algorithms utilizing a bag-of-words paradigm [26].

*Extractive summarization* is an NLP task whose objective is to generate a summary of a given text by selecting and combining important sentences or phrases from the original document [27]. The summary is created by extracting sentences that are considered the most informative of the original content. Various statistical and machine learning techniques have been developed over the years; good surveys of the field are given in [27,28]. Modern top-line techniques for extractive summarization rely on transformers [29,30].

Keywords are specific words or phrases that carry significant meaning or relevance within a particular context or subject matter. In the context of resume matching or text analysis, keywords are often used to represent essential skills, qualifications, experience, or specific terms related to a job or topic. When analyzing text, including resumes or job vacancies, keywords play a vital role in identifying and matching relevant information.

The field of *keyword extraction* focuses on the task of automatically identifying and extracting important keywords or key phrases from a given text. This process plays a crucial role in various natural language processing (NLP) applications, such as information retrieval, document summarization, text classification, and topic modeling. The methods for keyword extraction range from tf-idf-based [31,32] to graph-based, such as TextRank [33], to co-occurrence-based methods, such as RAKE [34], YAKE [35], and others [36,37]. Finally, neural methods have taken center stage during the last years [38,39].

A *named entity* refers to a specific real-world object or concept that has a proper name or designation. It can be an individual, organization, location, date, time, product, or any other entity that is referred to by a particular name or label. Recognizing and extracting named entities from text is an important task in natural language processing and information retrieval (IR). Methods for named entity recognition (NER) use probabilistic models such as conditional random fields [40–42], recurrent neural networks [43], transformers [44,45], graph neural networks [46,47], and transfer learning [48].

In this paper, we propose a method for efficient resume–vacancy matching by using summarization and enhancing the produced summaries by keywords or named entities found in texts. We use statistical and semantic vector representations for the enhanced texts and compute their similarity to produce a vacancy ranking for every resume. The method is unsupervised and does not require training. We perform an extensive experimental evaluation of our method to show its validity.

We address the following research questions in this paper:

- RQ1: does text enhancement with keywords or named entities improve ranking results?
- RQ2: does a summarization of either resumes or vacancies improve ranking results?
- RQ3: what text representation produces the best results?
- RQ4: can our method distinguish between vacancies relevant and irrelevant for an applicant?

This paper is organized as follows: Section 2 describes the details of our method—text processing and text types, text enhancement with keywords and named entities, and text representations. In Section 3, we describe the baselines to which we compare our method. Section 4 describes how our datasets were created, processed, and annotated. Metrics used for experimental evaluations and the rationale behind them are reported in Section 5. Section 6 provides full details of the experimental evaluation. Finally, Section 8 contains conclusions drawn from the evaluation.

## 2. Method

Our method is based on vector similarities and is therefore called Vector Matching (VM) method. We provide its outline and details in the subsections below.

### 2.1. The Outline

As a first step, we select one of the following text types for resumes and vacancies:

- Full texts;
- Summarized texts (summaries);
- Summaries enhanced by keywords;
- Summaries enhanced by named entities.

The extraction of summaries, keywords, and name entities is described below in Section 2.2.

In the second step, we represent the text data as numeric vectors with one of the methods described below in Section 2.3. Thus, for a resume, $R$, represented and a vacancy,

$V$, we have vectors $V_{resume}$ and $V_{vacancy}$. Then, we compute an L1 distance between a resume and a vacancy vector of length $n$:

$$\mathrm{L1}(V_{resume}, V_{vacancy}) = \sum_{i=1}^{n} \left| V_{resume}[i] - V_{vacancy}[i] \right|$$

The resulting ranking is inferred based on these distances, arranged in ascending order from the smallest to the largest. A pipeline of our approach is given in Figure 1.

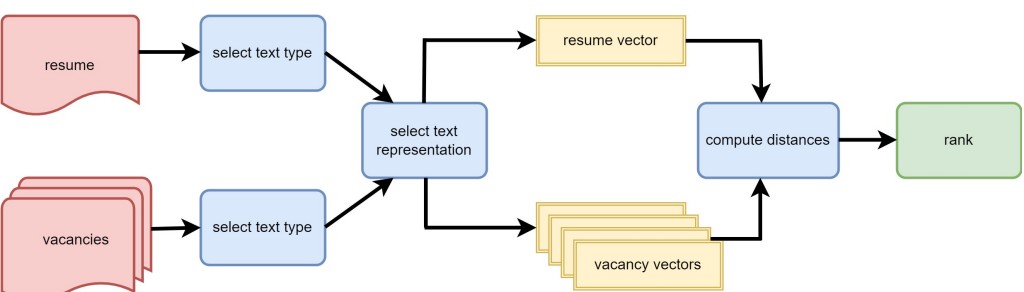

**Figure 1.** A pipeline of the VM method.

### 2.2. Text Types

In this section, we describe the text types that our ranking method uses. An overview of data representations is provided in Figure 2, and a detailed description is given in the sections below.

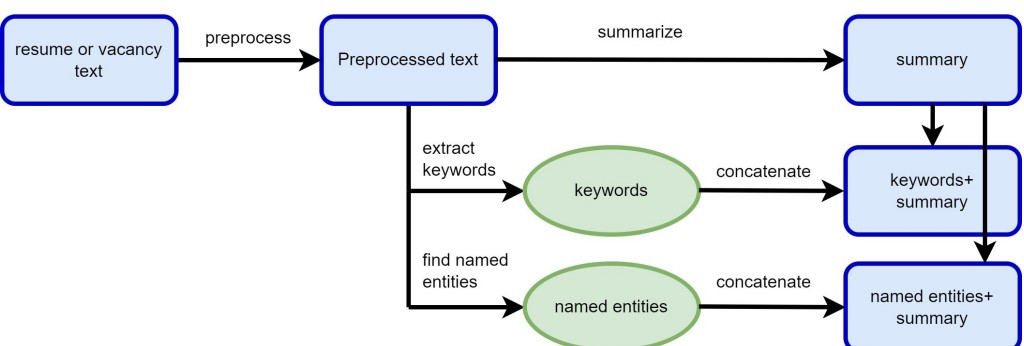

**Figure 2.** Text types used by the VM method.

### 2.2.1. Full Texts

As a first option, the VM method uses full texts of resumes and vacancies after basic preprocessing that includes removing numbers and converting them to lowercase.

### 2.2.2. Extractive Summaries

We use BERT-based extractive summarization [49] with a pre-trained English model `bert-base-uncased` to obtain summaries for vacancies and resumes. The method performs tokenization and then encodes tokens and feeds into a transformer that predicts the saliency scores of each sentence. The saliency scores indicate the importance or relevance of each sentence in the context of the overall document. We set the maximum size of a summary to 10 sentences, and the ELBOW method of BERT summarizer is used to determine the optimal summary length. The final summaries are computed from preprocessed full texts and contain at most 10 sentences.

### 2.2.3. Keyword-Enhanced Summaries

In the context of resume-matching systems, keywords from job vacancies are often compared with keywords extracted from resumes to determine the level of alignment between the candidate's profile and the job requirements. The presence or absence of

specific keywords can be used as a signal to assess the suitability of a candidate for a particular position [50].

To build keyword-enhanced summaries, we attach the keywords extracted from full texts to the summaries and pass them on to the text representation module. We do it because the extracted keywords may not be contained in a summary.

KeyBERT [39] is a method for keyword extraction that utilizes a BERT transformer-based language model to generate contextually relevant keywords. It aims to capture the semantic meaning of the text and identify key phrases that best represent the content. It employs a two-step procedure of embedding generation and keyword selection with a maximal marginal relevance (MMR) algorithm to select the most informative keywords from the generated embeddings. KeyBERT is a flexible method that can extract both single-word and multi-word keywords, and its implementation is available at https://github.com/MaartenGr/KeyBERT (accessed on 1 June 2023).

We applied KeyBERT and extracted multi-word keywords from resumes and vacancies with word numbers in the range $[1, 3]$. Examples of keywords for both types of data appear in Table 1, together with their KeyBERT scores.

**Table 1.** Keyword examples.

| Data Type | Keywords | Score |
| --- | --- | --- |
| Vacancy | Software development | 0.5359 |
| | Using software development | 0.529 |
| | Purpose development and | 0.5233 |
| | Purpose development | 0.5101 |
| | Maintenance of software | 0.5049 |
| Resume | Java backend development | 0.6156 |
| | Backend java developer | 0.6135 |
| | Java backend | 0.6066 |
| | Backend java | 0.5917 |
| | Now backend java | 0.58 |

2.2.4. Named-Entity-Enhanced Summaries

To build this text representation, we append the NEs extracted from full texts to the summaries and pass them on to the text representation module. The details of NE extraction are provided below.

We have used the spaCy SW package [51] to perform the NE extraction to preprocessed resumes and vacancies. By leveraging a pre-trained model, spaCy's named entity recognition (NER) module is capable of accurately recognizing and extracting named entities from text. The main types of spaCy NE are described in the Appendix A. We use the *en_core_web_sm* that is trained in English and, therefore, suits our data. Examples of NEs extracted from the data appear in Appendix A.

To construct data representations, we append named entities to the texts, whether in their complete form or as summaries, and then feed them into the text representation module.

*2.3. Text Representation*

Our vector matching method uses the following text representations:

1. Word *n*-grams with $1 \leq n \leq 3$. A word *n*-gram is a contiguous sequence of *n* words within a text or a sentence.
2. Character *n*-grams with $1 \leq n \leq 3$. A character *n*-gram is a contiguous sequence of *n* characters within a text.
3. BOW vectors with TF-IDF weights, where every resume and vacancy is treated as a separate document. A BOW vector is a numerical representation of a text document that captures the occurrence or frequency of words in the document.

4. Bidirectional encoder representations from transformers (BERT) sentence embeddings obtained from a pre-trained English BERT model "bert-base-uncased" [52]; this model was selected because all the resumes are written in English.

These representations are computed for every one of the text data forms described in Section 2.2.

## 3. Baselines

### 3.1. The OKAPI Algorithm

The Okapi BM25 (Best Matching 25) algorithm [53] is a ranking function commonly used in information retrieval systems to calculate the relevance of a document to a given query. It is an improved version of the original Okapi BM11 algorithm [24].

The algorithm takes into account factors such as term frequency, document length, and the overall distribution of terms in a collection of documents. It aims to estimate the probability that a document is relevant to a particular query. The algorithm performs tokenization and computes a score for a document–query pair using the following components: term frequency and inverse document frequency for every term (token), document length, query length, average document length, and two free parameters, $k1$ and $b$, which control the impact of term frequency and document length normalization, respectively. The OKAPI-BM25 algorithm was shown to be a useful baseline for experiments and features for ranking [26].

### 3.2. BERT-Based Ranking

We use the method introduced in [54] as a neural baseline, which we denote by BERT-rank. This method applies a knowledge distillation approach that transfers knowledge from a larger and more accurate teacher model to a smaller student model. The knowledge transfer is performed across different neural architectures, allowing the student model to benefit from the teacher model's insights. The proposed method employs multiple techniques, including soft target training, attention alignment, and feature distillation, to effectively transfer knowledge.

## 4. The Data

### 4.1. Vacancies Dataset

We downloaded job postings titled "software developer" in the United States during August 2019 [55]. These data were extracted using JobsPikr—a job data delivery platform that extracts job data from job boards across the globe [56]. JobsPikr crawls over 70,000 employer websites daily to process over one million job postings; its API provides access to a database of over 500 million jobs over the past 3 years [57]. The data were downloaded in .csv format, and they contain 10,000 vacancies with a combined size of 41.51 MB. The following data issues were found: (1) the same vacancy can appear more than once with a different unique id, and (2) portions of text in the job description field are repeated in many cases.

An example of a vacancy and dataset statistics are provided in Appendix A.

We extracted the names of vacancies using the "find-job-titles" library [58]. To find vacancies relevant to our applicants we select those that include the word "developer" or "full-stack" from the entire array of vacancies. Table 2 describes the column and context statistics of the data, providing the minimum, maximum, and average character length for every field, data type, and the number of non-empty entries for each field. Table 3 gives an example of a relevant vacancy as it appears in the vacancies dataset.

To find vacancies that are not relevant to our applicants, we downloaded them with JobsPikr [56] from the *"10,000-data-scientist-job-postings-from-the-usa"* directory (a partial example of such a vacancy is given in Table 4). Table 5 shows examples of NE and keywords that are extracted from this type of vacancy; while not all vacancies in this set are for data scientists, they are intended for people looking for senior positions only.

**Table 2.** Vacancy Dataset Features.

| Field Name | Type | Non-Empty Count | Min Length | Max Length | Avg Length |
|---|---|---|---|---|---|
| Crawl_timestamp | String | 10,000 | 25 | 25 | 25 |
| URL | URL | 10,000 | - | - | - |
| Job_title | String | 10,000 | 18 | 14,230.102 | |
| Category | String | 9161 | 2 | 38 | 18.828 |
| Company_name | String | 10,000 | 2 | 277 | 27.285 |
| City string | 9869 | 1 | 41 | 10.141 | |
| Country | String | 10,000 | 2 | 3 | 2.559 |
| Inferred_city | String | 7829 | 3 | 22 | 8.649 |
| Inferred_state | String | 9207 | 4 | 20 | 8.325 |
| Inferred_country | String | 9229 | 3 | 13 | 8.602 |
| Post_date | Date | 10,000 | - | - | - |
| Job_description | String | 10,000 | 64 | 27,576 | 3966.576 |
| Job_type | String | 10,000 | 8 | 10 | 8.938 |
| Salary_offered | String | 0 | - | - | - |
| Job_board | String | 10,000 | 4 | 13 | 7.551 |
| Geo string | 9999 | 3 | 3 | 3 | |
| Cursor | Integer | 10,000 | - | - | - |
| Contact_email | String | 0 | - | - | - |
| Contact_phone_number | String | 819 | 1 | 60 | 12.009 |
| Uniq_id | String | 10,000 | 32 | 32 | 32 |

**Table 3.** A sample developer vacancy from the vacancies dataset.

| Column Name | Content |
|---|---|
| Crawl_timestamp | 6 February 2019 05:53:38 +0000 |
| URL | https://www.dice.com/jobs/detail/C%252B%252B-Software-Developer-SigmaWay-San-Jose-CA-95101/90994229/801699 |
| Job_title | C++ Software Developer |
| Category | Computer-or-internet |
| Company_name | SigmaWay |
| City | San Jose |
| State | CA |
| Country | USA |
| Inferred_city | San Jose |
| Inferred_state | California |
| Inferred_country | USA |
| Post_date | 5 February 2019 |
| Job_description | Apply by Email/Direct Application at udit.sharma@sigmaway.org C++ Software Developer Duration: 12+ Months Location: Sunnyvale, CA Top 3 Must Haves 1. C++ Software Experience (5+ years) 2. Education in information technology, computer science, or related field. Masters is preferred. 3. Knowledge of Software Development and Design across multiple platforms Preferred but not required Having experience with robotics projects or systems is a plus Required Qualifications Bachelors degree in information technology, computer science, or related field. Masters is preferred. At least 5 years experience in C++ development Excellent knowledge in algorithms and data structures Experienced in professional software development using versioning systems (git), reviewing systems Experience in agile development projects Excellent teaming and communication skills Ability to work in cross-functional teams |
| Job_type | Undefined |
| Salary_offered | |
| Job_board | Dice |
| Geo | USA |
| Cursor | 1,549,436,429,136,812 |
| Contact_email | Empty 100% |
| Contact_phone_number | (408) 627-7905 |
| Uniq_id | 3b22ba3aa471681a909f878d8cec1b58 |

**Table 4.** An example of a data science vacancy.

| Field | Content |
|---|---|
| Job_description | Data Scientist EnvoyIT is looking for a Data Scientist for a Full-time position with one of our Healthcare clients in Reston, VA You will be joining a new initiative. This is an outstanding employer with plenty of growth opportunity. Base plus bonus and benefits. PURPOSE: As a Data Scientist, you will: Join a brand new team of machine learning researchers with an extensive track record in both academia and industry. Bring a combination of mathematical rigor and innovative algorithm design to create recipes that extract relevant insights from billions of rows of data to effectively & efficiently improve health outcomes. Create thoughtful solutions that engage and empower members to make more informed decisions about their health Develop statistical applications that can be reproduced and deployed on enterprise platforms. Develop functional means for measuring the quality of healthcare members receive annually. Interact with and report to an audience that includes Directors, Vice-Presidents and the C-level executives. Build tools and support structures needed to analyze data, perform elements of data cleaning, feature selection and feature engineering and organize experiments in conjunction with best practices. Assess the effectiveness and accuracy of new data sources and data gathering techniques. |
| Job_title | Data Scientist |
| Uniq_id | eda91b88eb3096ed98bc1a5f6b5568df |

**Table 5.** Keyword and NE examples for a data science vacancy.

| Keyword | Score | Named Entity | Type |
| --- | --- | --- | --- |
| Synergy software design | 0.6431 | 3d | CARDINAL |
| Development team synergy | 0.5812 | 2004 | DATE |
| Synergy software | 0.5392 | Each | DATE |
| Com synergy software | 0.5264 | Year | DATE |
| Team synergy software | 0.5223 | Maritime | FAC |
| | | Plaza | FAC |

*4.2. Resume Dataset*

As we did not succeed in finding a good-quality resume dataset, we constructed our own by downloading real resumes from the Telegram [59] group "HighTech Israel Jobs" [60], which freely publishes resumes and vacancies.

All resumes in the group are written in a free format. Most often they consist of one page, but some have two or more pages. There are no clear structures, marks, or mandatory sections. In the original dataset, there are summaries in Russian and English; it happens that one person uploads several versions of their resume. The following data preprocessing steps were made:

1.　Exclusion of duplicate records from the dataset.

We use the "hashlib" library [61] that determines how many bytes should be checked. Then, the function recursively checks all the specified paths. The output is duplicated addresses.

2.　Exclusion of resumes not written in English.

We use the "detect" function of the "langdetect" library [62]. The contents of the files were extracted and checked. This function returns the name of the language of the text. If it returned "en", which is English, then we accept this file; otherwise, we delete it.

3.　Exclusion of resumes of non-developers.

More than 90% of the resumes were eliminated at the third stage because the dataset contained a lot of resumes of testers, project managers, etc. We used a custom-made list of keywords to search for in candidates' resumes using the "flashtext" library [63]. A sample of the list containing job titles for Oracle developers is provided in Table 6.

4.　Anonymization.

Even though the resumes are published in an open Telegram group, we masked the names, phone numbers, and addresses of applicants.

Figure 3 gives an example of a resume from this dataset.

*4.3. Human Annotation*

Human rankings are essential for evaluating resume ranking systems because they provide a crucial benchmark for measuring the effectiveness and accuracy of automated resume–vacancy matching methods. To produce these rankings, we use the following pipeline:

1.　First, we selected a subset of the resume dataset containing 30 resumes (vacancy rankings are time-consuming, and the availability of our annotators was limited).
2.　Then, we selected a random subset of the vacancies dataset containing five relevant vacancies.
3.　Two annotators with a computer science background were asked to rank the vacancies from the most relevant (rank 1) to the least relevant (rank 5) for every one of the 30 resumes.

　　Our annotators received detailed guidelines from a manager of recruitment, organizational development, and welfare in an HR department of a large academic institution. Mrs. Yaarit Katzav (https://en.sce.ac.il/administration/human-resources1/staff (accessed on 1 June 2023)) is a senior HR manager with over 13 years of experience who is responsible, among other things, for recruiting information systems engineers

at SCE Academic College. The full vacancy–resume matching guidelines provided by her appear in Table 7.

4. Finally, we received 30 rank arrays of length 5 from both of our annotators and computed an average rank array of length 5 for every resume. (The resume dataset and annotations are freely available at https://github.com/NataliaVanetik/vacancy-resume-matching-dataset).

We use these human rankings as the ground truth for evaluating the performance of our method and competing methods; we compare the ground-truth rankings with the rankings produced by automatic methods for the selected 30 resumes and 5 vacancies.

---

**XXXXXXXXXXXXXXX**
**Java full stack developer**
**(60% – IN JAVA BACKEND DEVELOPMENT AND 40% – IN WEB FRONTEND)**
3+ years of experience FullStack development and 5+ total experience in software development

**phone** xxxxxx                                    **linkedIn** xxxxxx
**location** Bat Yam                            **email** xxxxxx

**Professional Skills**

Programming languages: Java, JavaScript.
Web:  HTML5/CSS3, JQuery, Bootstrap3
Environments – IDE/tools:  Eclipse, IntelliJ IDEA, GitHub.
Databases: SQL (MySQL), NoSQL (MongoDB).
Technologies & Frameworks: OOP, AOP, Spring MVC, JPA, Hibernate, JDBC, SpringBoot, Spring Security, Spring Web, REST, JSON, Maven, Junit, Git, Postman.
Systems: Linux, Windows.

**Professional Experience**

**2020 -  now:  Backend JAVA developer**,  Elpisor LTD (Israel, Rehovot)
**Latest project:** Participation in the development of a microservice architecture for a system for storing educational information.
**Areas of responsibility:**
- Developed backend website application - RESTful Web service (Spring Boot Starter Web);
- Authentication/Authorization (Spring Security);
- Working with MongoDB, MySQL;
- Writing and running junit tests, fixing defects
- Email feedback.
- JSON web token for user authentication using Swagger UI to render and interact with API, REST, HTTPS.
**Tools summary:** Spring Boot, Spring Security, Hibernate ORM, JSON, REST, MySQL, MongoDB, Java Persistence API, Maven, JDBC, Apache Tomcat, JUnit, HTML5, CSS3.

**2017 – 2019:  Full stack JAVA developer**, Bank Otkritie (Russia, Moscow)
- Preparation of non-standard reports by executing queries to SQL databases. Creating a simple web interface for entering a request and getting the result in a user-friendly form        (HTML, CCS, Bootstrap, JavaScript).
- Performed hardware and software installations and provided high-level customer care, training, and technical support.

**2005 - 2017: Software R-Style language developer,** Privatbank (Ukraine, Dnepr)
- Performed hardware and software installations and provided high-level customer care, training, and technical support.
- Introduction of changes in the terms of service for bank products into the business logic of the program code.(R-Style Language).

**Education**
**2000 - 2005: master's degree in Computer Science and Information Technology.**
**Specialization: design and production of electronic computing systems.**
National Technical University of Ukraine

**Other**
**Languages:** English, Hebrew, Russian, Ukrainian.
**Recommendations are available upon request**

**Figure 3.** Example of a resume from the resume dataset.

**Table 6.** A list of job titles for Oracle developers.

| Job Title |
| --- |
| Oracle ADF Developer, Oracle Apex Developer, Oracle Applications Developer, Oracle BPM Developer, Oracle BRM Developer, Oracle Business Intelligence Developer, Oracle Data Warehouse Developer, Oracle Database Developer, Oracle Developer, Oracle E Business Developer, Oracle EBS Developer, Oracle ERP Developer, Oracle ETL Developer, Oracle Financial Application Developer, Oracle Financials Developer, Oracle Forms Developer, Oracle Fusion Developer, Oracle Fusion Middleware Developer, Oracle HRMS Developer, Oracle OBIEE Developer, Oracle PL SQL Developer, Oracle R12 Developer, Oracle Reports Developer, Oracle SOA Developer, Oracle SQL Developer, Oracle Technical Developer |

**Table 7.** Ranking guidelines.

| Parameter | Explanation |
| --- | --- |
| Work experience | Candidates should ensure that their work experience matches the level required for the job postings they are applying for. |
| Qualifications | Educational and professional qualifications are also important considerations. Candidates should make sure that they have the necessary qualifications required for the job. |
| Clarity | Candidates should look for job postings with clear and specific job requirements and job culture descriptions, and avoid applying for roles with vague or generic requirements. They should tailor their applications to the specific job posting, highlighting the most important requirements that match their skills and qualifications. |
| Conciseness | Some job postings might have a long list of requirements. Try to focus on the most important requirements that match your skills and qualifications. |
| Keywords | Candidates should focus on the most relevant keywords and requirements that match their skills and qualifications, and avoid irrelevant information. |
| Industry trends | Candidates should stay up to date with the latest industry trends and developments in the IT field. This can help them identify new job opportunities and position themselves as experts in their field. |

## 5. Metrics

To analyze the correlation between ground-truth ratings and ratings produced by evaluated methods, we consider several metrics that are suited for comparing rankings.

Krippendorff's alpha [64] is a metric used for assessing the agreement or reliability of rankings or ordinal data among multiple raters or annotators. The metric takes into account both the observed agreement and the expected agreement between raters, providing a measure of inter-rater reliability. It considers the differences between ranks assigned by different raters and compares them to the expected differences based on chance:

$$\alpha = 1 - \frac{\text{expected agreement}}{\text{observed agreement}}$$

The metric values range from $-1$ to 1, with higher values indicating greater agreement among raters.

Spearman's rank correlation coefficient [65] is a statistical measure used to assess the strength and direction of the monotonic relationship between two ranked variables. When applied to ranking comparison, Spearman's rank correlation coefficient quantifies the similarity or agreement between two sets of rankings. It measures how consistently the orderings of the items in the two rankings correspond. It is computed as

$$\rho = 1 - \frac{6 \sum d_i^2}{n(n^2 - 1)}$$

where $\rho$ is Spearman's rank correlation coefficient, $d_i$ stands for the difference between the ranks of the corresponding pair of items, and $n$ is the number of items or observations in each ranking.

Cohen's kappa coefficient [66] is a statistical measure used to assess the agreement between two raters or annotators when working with categorical or ordinal data. When applied to a comparison of rankings, Cohen's kappa coefficient quantifies the agreement between two sets of rankings, considering the chance agreement that could occur by random chance. It is computed as

$$\kappa = \frac{\text{observed agreement} - \text{expected agreement}}{1 - \text{expected agreement}}$$

From our observations, the three metrics correlated perfectly in all our experiments (i.e., produced the same rankings of methods). However, Krippendorff's alpha consistently displayed higher values than Cohen's kappa and Spearman's correlation. It can offer a more thorough measure of agreement and is made to handle situations with more than two raters or coders [67]. Cohen's kappa focuses on the pairwise agreement between two raters and may be less suggestive when comparing rankings. Therefore, we chose not to report Cohen's kappa values. We used Krippendorff's alpha and Spearman's correlation to evaluate the agreement among the annotators and the agreement of vacancy ranks produced by different systems with the ground truth.

## 6. Experimental Evaluation

### 6.1. Hardware and Software Setup

All experiments were performed on Google Colab [68] with Pro settings. We used spaCy [51] for sentence splitting, tokenization, and NE extraction. Sklearn [69] and SciPy [70] Python packages were used to compute evaluation metrics and text representations. For keyword extraction, we used the KeyBERT SW package [39] with default settings.

### 6.2. Methods

Our baselines for comparison are the original OKAPI BM25 algorithm [24] and the BERT-based method of [54] denoted by BERT-rank. Our vector matching method is denoted by VM, and we use all varieties of text data (described in Section 2.2) and text representation vectors (described in Section 2.3). We have a total of 4 text data formats (full texts, summaries, and summaries extended with either keywords or named entities) for both resumes and vacancies and 5 text representations (word and character $n$-grams, BERT sentence embeddings, tf-idf vectors, and concatenations of all of these vectors), which gives us a total of 80 options. Therefore, we only report the best-performing combinations in each case.

### 6.3. Metrics

We compare the rankings produced by every method to the ground-truth ranking and report Krippendorff's alpha (computed with ReCal web service [71]) and Spearman's correlation values for all of them. For the automatic tests, we also report accuracy (a full explanation is provided in Section 6.5).

### 6.4. Human-Annotated Data Evaluation

In these tests, we perform a ranking of 5 selected vacancies for all of the 30 resumes annotated by humans. The rankings are compared to the average ranking produced by our two human annotators. Table 8 shows the results of ranking our VM method and the results for our two baselines—OKAPI BM25 and BERT-based rank—as well. The best scores are marked with a gray background.

We only list the VM methods (sorted by decreasing Krippendorff's alpha) that outperform both baselines. The difference between the top method and baselines is very significant. Furthermore, we see that the best method uses *the full text of resumes and summarized vacancies and character n-grams as text representation*.

Figure 4 shows Krippendorff's alpha values for all text representations for the best text type setup (full texts for resumes and summaries for vacancies). We observe that the best scores are achieved by the two representations that incorporate character $n$-grams and that the difference from other text representations is significant. Because pre-trained BERT models are trained on generic texts rather than resumes and vacancies, BERT sentence vectors perform poorly as text representation. The tf-idf vectors also underperform because vacancies and resumes come from a variety of candidates and firms, and less significant terms may be given greater weights simply because they are uncommon, rather than because they are important.

We can also see that the top three methods use summarized texts for either vacancies or resumes or both. Indeed, from looking at the texts, we can see that vacancy descriptions often contain excessive information, and resumes contain parts that are too generic. Our HR consultant has confirmed that the same resume is often submitted to several positions, and it is not adapted to a specific vacancy. She also explained that, in most cases, only parts of the vacancy description are specific to the job, and the rest often contains general company information.

Figure 5 shows the behavior of the VM method when the same text type is chosen for both resumes and vacancies. We can see that character $n$-gram-based text representation provides the best performance regardless of what text type (full text, summary, or

keyword- and named-entity-enhanced summaries) was chosen, and that tf-idf gives the worst performance in all cases.

We have tested the possibility of using cosine similarity for this task, but it resulted in inferior performance. It turns out that the L1 distance is more suitable for this specific task. The intuition for this phenomenon is that cosine similarity treats all features equally, while L1 distance allows for feature-specific weighting, enabling the prioritization of specific features in the similarity calculation.

**Table 8.** Evaluation results for baselines and the VM method on human-annotated data.

| Baseline | | | | Krippendorff's Alpha | Spearman's Correlation |
|---|---|---|---|---|---|
| OKAPI-BM25 | | | | 0.3055 | 0.2262 |
| BERT-rank | | | | −0.1779 | −0.3071 |
| Method | Resume text data | Vacancy text data | Text representation | | |
| VM | Full text | Summary | Char ng | 0.6287 | 0.5908 |
| VM | Full text | Summary | tfidf + char ng + word ng + sbert | 0.6179 | 0.5794 |
| VM | Full text | kw + summary | Char ng | 0.5885 | 0.5455 |
| VM | Full text | kw + summary | tfidf + char ng + word ng + sbert | 0.5075 | 0.4555 |
| VM | kw + summary | Full text | tfidf + char ng + word ng + sbert | 0.4918 | 0.4444 |
| VM | kw + summary | Full text | Char ng | 0.4918 | 0.4444 |
| VM | ne + summary | Full text | tfidf + char ng + word ng + sbert | 0.4918 | 0.4444 |
| VM | ne + summary | Full text | Char ng | 0.4918 | 0.4444 |
| VM | Summary | Full text | tfidf + char ng + word ng + sbert | 0.4918 | 0.4444 |
| VM | Summary | Full text | Char ng | 0.4918 | 0.4444 |
| VM | ne + summary | Full text | tfidf | 0.4858 | 0.4377 |
| VM | Full text | Full text | tfidf | 0.4776 | 0.4273 |
| VM | Full text | Full text | tfidf + char ng + word ng + sbert | 0.4743 | 0.4210 |
| VM | kw + summary | Full text | tfidf | 0.4716 | 0.4206 |
| VM | Summary | Full text | tfidf | 0.4716 | 0.4206 |
| VM | Full text | Full text | Char ng | 0.4697 | 0.4192 |
| VM | kw + summary | Summary | sbert | 0.3695 | 0.3003 |
| VM | ne + summary | ne + summary | Char ng | 0.3645 | 0.2972 |
| VM | kw + summary | ne + summary | tfidf + char ng + word ng + sbert | 0.3623 | 0.2892 |
| VM | kw + summary | Full text | Word ng | 0.3552 | 0.2900 |
| VM | ne + summary | Full text | Word ng | 0.3552 | 0.2900 |
| VM | Summary | Full text | Word ng | 0.3552 | 0.2900 |
| VM | Full text | Full text | Word ng | 0.3552 | 0.2900 |
| VM | ne + summary | Full text | sbert | 0.3466 | 0.2773 |
| VM | Summary | ne + summary | Char ng | 0.3383 | 0.2625 |
| VM | Summary | ne + summary | tfidf + char ng + word ng + sbert | 0.3312 | 0.2556 |
| VM | ne + summary | Summary | sbert | 0.3155 | 0.2403 |
| VM | kw + summary | ne + summary | Char ng | 0.3154 | 0.2377 |
| VM | Full text | Full text | sbert | 0.3097 | 0.2340 |

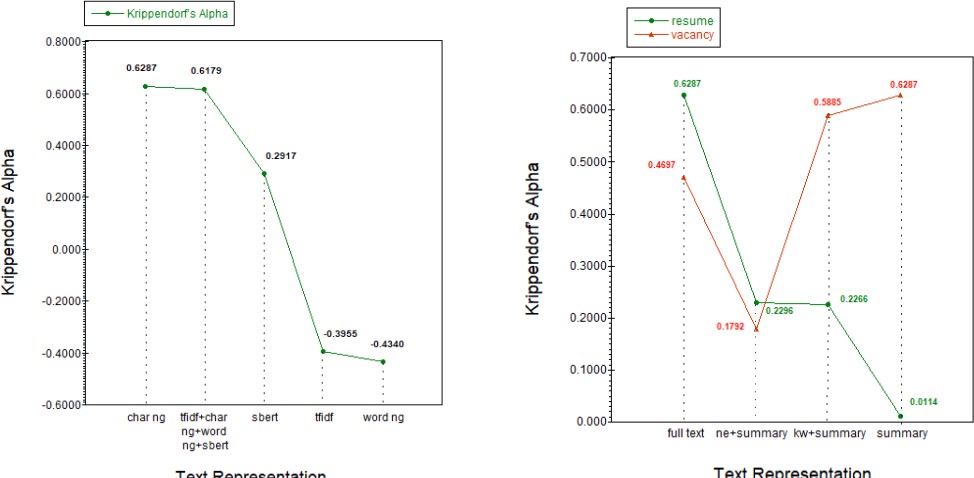

**Figure 4.** Effect of text representations (**left**) and text data choice (**right**) on the best model configuration (dotted lines indicate text representations.)

In Table 9, we list the scores of the top 10 variations in the VM method (according to their Krippendorff's alpha value) when cosine similarity is used. The best scores are marked with a gray background color. Although most of these scores are still better than the baselines' results, their values are significantly lower than in Table 8. We do, however, observe that text representations that use character *n*-grams produce the best results in this setup as well.

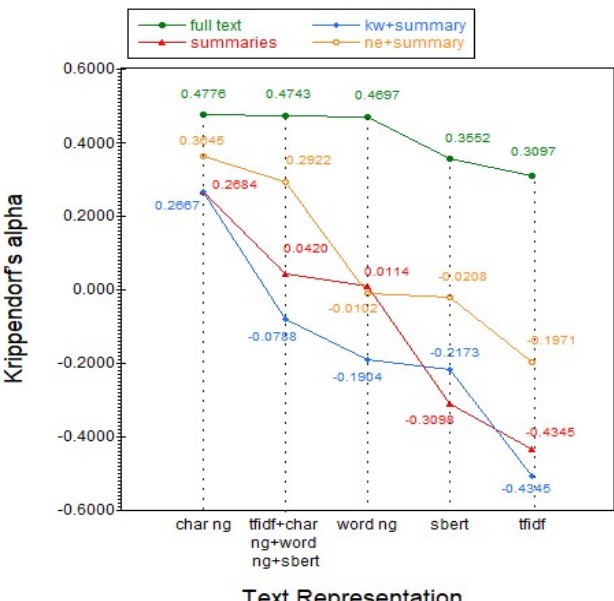

**Figure 5.** Choosing the same text type for resumes and vacancies (dotted lines indicate text types.)

**Table 9.** Results for the VM method with cosine similarity on human-annotated data.

| Method | Resume Text Data | Vacancy Text Data | Text Representation | Krippendorff's Alpha | Spearman's Correlation |
|---|---|---|---|---|---|
| VM cosine | Full text | Full text | Char ng | 0.5186 | 0.4708 |
| VM cosine | Full text | Full text | tfidf + char ng + word ng + sbert | 0.5006 | 0.4508 |
| VM cosine | Summary | Full text | tfidf + char ng + word ng + sbert | 0.4473 | 0.3894 |
| VM cosine | Summary | Full text | Char ng | 0.3993 | 0.3361 |
| VM cosine | ne + summary | Full text | tfidf + char ng + word ng + sbert | 0.3741 | 0.3021 |
| VM cosine | ner + summary | Full text | Char ng | 0.3681 | 0.2955 |
| VM cosine | kw + summary | Full text | tfidf + char ng + word ng + sbert | 0.3216 | 0.2516 |
| VM cosine | kw + summary | Full text | Char ng | 0.3216 | 0.2516 |
| VM cosine | ner + summary | ner + summary | sbert | 0.2612 | 0.1867 |
| VM cosine | Full text | ner + summary | sbert | 0.1892 | 0.1067 |
| VM cosine | Summary | ner + summary | sbert | 0.1719 | 0.0861 |

We also performed an additional human evaluation for the relevance of results produced by the top competing model (OKAPI-BM25) and our top model (VM with full-text resumes, summarized vacancies, and character n-gram text representation). The top-ranked job openings selected by the two top techniques for each of the 66 resumes in the whole dataset were provided to the annotators. Every resume–vacancy pair was given a relevance score from 1 to 3, with 1 denoting that the position is unimportant, 2 denoting that it is somewhat relevant, and 3 denoting that the position is relevant to the applicant. Each resume and vacancy was assessed by two different annotators. Average relevancy scores for the two methods appear in Table 10. We can see that the best vacancies selected by the top VM method are more relevant than those selected by the OKAPI-BM25 algorithm. A pairwise two-tailed statistical significance test [72] showed that the difference in assessments for these methods is statistically significant with a *p*-value of 0.0455.

**Table 10.** Relevance evaluation for the top VM method and OKAPI-BM25.

| Method | Average Relevance |
|---|---|
| OKAPI-BM-25 | 2.0227 |
| VM CV full text + vacancy summary + char ng | 2.1894 |

### 6.5. Automatically Annotated Data Evaluation

To be less dependent on human annotations, we have designed the following test that can be applied to all resumes in our dataset.

We have extracted one random relevant developer vacancy, $D_{\text{developer}}$, from our dataset and one data scientist/senior vacancy, $D_{\text{data scientist}}$. Because our resume dataset contains data for software developers and full-stack engineers only, this type of vacancy is not relevant for them. Therefore, the ground-truth ranking for all resumes has the form

$$[D_{\text{developer}}, D_{\text{data scientist}}] \tag{1}$$

to which we can compare the rankings produced by our and baseline methods. In this case, we can report binary classification accuracy as well because we consider the answer (1) correct and the answer $[D_{\text{data scientist}}, D_{\text{developer}}]$ to be incorrect. Table 11 contains the results for baseline methods and all variations in the VM method that demonstrate non-negative Krippendorff's alpha values. The best scores are marked with a gray background. We can see that the top 5 VM methods demonstrate better performance than any of our baselines in all metrics, including accuracy. We can see that scores of all the methods improve, meaning that this is an easier task than the ranking of five relevant vacancies. The best performance is achieved when NE and summary are used as text data and character $n$-grams are chosen as text representation.

**Table 11.** Baselines and VM method results for automatically annotated data.

| Baseline | | | | Krippendorff's Alpha | Spearman's Correlation | Acc |
|---|---|---|---|---|---|---|
| OKAPI-BM25 | | | | 0.7000 | 0.6000 | 0.8000 |
| BERT-rank | | | | 0.2700 | 0.0300 | 0.5200 |
| Method | Resume text data | Vacancy text data | Text representation | | | |
| VM | Full text | ne + summary | Char ng | 0.9091 | 0.8788 | 0.9394 |
| VM | Full text | ne + summary | tfidf + char ng + word ng + sbert | 0.8636 | 0.8182 | 0.9091 |
| VM | Summary | Summary | Word ng | 0.8409 | 0.7879 | 0.8939 |
| VM | ne + summary | Summary | Word ng | 0.7955 | 0.7273 | 0.8636 |
| VM | kw + summary | Summary | Word ng | 0.7727 | 0.6970 | 0.8485 |
| VM | Full text | kw + Summary | Char ng | 0.6591 | 0.5455 | 0.7727 |
| VM | Full text | Summary | Char ng | 0.5455 | 0.3939 | 0.6970 |
| VM | Full text | kw + summary | tfidf + char ng + word ng + sbert | 0.5227 | 0.3636 | 0.6818 |
| VM | Full text | Summary | Word ng | 0.4091 | 0.2121 | 0.6061 |
| VM | Full text | Summary | tfidf + char ng + word ng + sbert | 0.3864 | 0.1818 | 0.5909 |
| VM | Summary | kw + summary | Word ng | 0.3636 | 0.1515 | 0.5758 |
| VM | ne + summary | kw + summary | Word ng | 0.3182 | 0.0909 | 0.5455 |
| VM | ne + summary | ne + summary | Word ng | 0.2500 | 0.0000 | 0.5000 |
| VM | ne + summary | ne + summary | sbert | 0.1818 | −0.0909 | 0.4545 |
| VM | Summary | ne + summary | sbert | 0.1818 | −0.0909 | 0.4545 |
| VM | Summary | ne + summary | Word ng | 0.1591 | −0.1212 | 0.4394 |
| VM | kw + summary | kw + summary | Word ng | 0.0909 | −0.2121 | 0.3939 |
| VM | Full text | ne + summary | sbert | 0.0227 | −0.3030 | 0.3485 |

### 6.6. Extended Human-Annotated Data Evaluation

Because human ranking is a time-demanding task, we suggest the following semi-automated test to further assess ranking methods. We expand the five annotated vacancies with a single random data scientist vacancy from JobsPikr to obtain six vacancies. Because the extra vacancy is not relevant for applicants in our 30 annotated resumes, its rank should be last for every single one of them. With this approach, we do not need to compute a new ground truth. Table 12 shows the results of ranking our VM method and the results for our two baselines—OKAPI BM25 and BERT-based rank—as well. The best scores are marked with a gray background. We only list the VM methods that outperform both baselines. We can see that VM methods perform much better than baselines and that the top method is

the same configuration that produced the best scores in Table 8. Furthermore, we see that OKAPI performs significantly better when there is one irrelevant in the list than in the case where there is none (Table 8).

**Table 12.** Baselines and VM method results for extended human-annotated data.

| Baseline | | | | Krippendorff's Alpha | Spearman's Correlation |
|---|---|---|---|---|---|
| OKAPI-BM25 | | | | 0.5350 | 0.4908 |
| BERT-rank | | | | −0.2905 | −0.4070 |
| Method | Resume text data | Vacancy text data | Text representation | | |
| VM | Full text | Summary | Char ng | 0.7630 | 0.7435 |
| VM | Full text | Summary | tfidf + char ng + word ng + sbert | 0.7411 | 0.7198 |
| VM | Full text | kw + summary | Char ng | 0.7378 | 0.7158 |
| VM | kw + summary | Full text | tfidf + char ng + word ng + sbert | 0.7045 | 0.6827 |
| VM | kw + summary | Full text | Char ng | 0.7045 | 0.6827 |
| VM | ne + summary | Full text | tfidf + char ng + word ng + sbert | 0.7045 | 0.6827 |
| VM | ne + summary | Full text | Char ng | 0.7045 | 0.6827 |
| VM | Summary | Full text | tfidf + char ng + word ng + sbert | 0.7045 | 0.6827 |
| VM | Summary | Full text | Char ng | 0.7045 | 0.6827 |
| VM | Full text | kw + summary | tfidf + char ng + word ng + sbert | 0.6877 | 0.6605 |
| VM | kw + summary | Full text | Word ng | 0.6252 | 0.5947 |
| VM | ne + summary | Full text | Word ng | 0.6252 | 0.5947 |
| VM | Summary | Full text | Word ng | 0.6252 | 0.5947 |
| VM | Full text | Full text | Word ng | 0.6252 | 0.5947 |
| VM | Full text | Full text | tfidf + char ng + word ng + sbert | 0.5976 | 0.5643 |
| VM | Full text | Full text | Char ng | 0.5648 | 0.5298 |
| VM | kw + summary | Summary | sbert | 0.5587 | 0.5185 |
| VM | Full text | Summary | sbert | 0.5375 | 0.4958 |

## 7. Limitations

### 7.1. Resume Type

The proposed method focuses solely on information technology (IT) resumes and vacancies due to the data availability for this specific domain. As a result, the findings and conclusions drawn in this study are limited to the context of IT-related job applications and may not be directly applicable to other industries or professions. The exclusion of data from other fields may restrict the generalizability of the research outcomes beyond the IT domain. Our future research will try to combine diverse datasets from other areas in order to provide a more thorough understanding of automatic resume–vacancy matching systems in various job sectors.

### 7.2. Resume Structure

The vacancy–resume matching method used in this study is limited to unstructured resumes alone, thereby limiting its application. Unstructured resumes lack a uniform format and can vary greatly in content and organization, making it difficult to reliably extract useful information; while unstructured resumes are common, many firms also receive structured resumes via online application portals where individuals enter their information into predetermined fields. However, more study and methodological changes would be required to broaden the application of our matching methodology.

## 8. Conclusions

In this paper, we presented a new annotated resume dataset for the task of a resume–vacancy ranking. Additionally, we have proposed a method based on the statistical and semantic vector representation of texts constructed with NLP techniques. The method is unsupervised and does not require training. We compared our VM method with two baselines—OKAPI BM25 and BERT-rank—using various text data formats and representations. The results of our evaluation demonstrate that our method outperforms both baselines in terms of ranking accuracy. Specifically, we found that using the full text of resumes and summarized vacancies, along with character-n-gram-based text representation, yielded the best performance. This combination achieved a significantly higher Krippendorff's alpha value compared to the baselines. Furthermore, we analyzed the impact of different text

representations and text data types on the performance of our VM method. We observed that text representations incorporating character *n*-grams consistently produced the best results, while tf-idf-based representations yielded the lowest performance.

Additionally, we found that using the same text type for both resumes and vacancies, particularly character-n-gram-based representations, resulted in improved rankings. It also became apparent that the L1 distance was better suited for this specific ranking task; while the performance using cosine similarity surpassed the baselines, the scores were noticeably lower when compared to the L1-distance-based approach.

Overall, the summarization of vacancies does improve the ranking results; however, it is not the best text type choice for the resumes. Therefore, the answer to research question 2 is partially true. However, adding keywords does not produce the best results except for the case of automatic evaluation where an irrelevant vacancy is present. The answer to research question 1 concerning named entities depends on the data—we only benefit from named entities if we have vacancies unrelated to our applicants. We have found that the best text representations in every setup are the ones that use character *n*-grams, thus answering research question 3. Automatic and semi-automatic evaluations in Sections 6.5 and 6.6 show that the performance of all the methods, including OKAPI, improves when we evaluate resumes against vacancies that are not relevant for the applicants. Moreover, the advantage of our approach is preserved in these cases too. Therefore, the answer to research question 4 is positive.

**Author Contributions:** Conceptualization, N.V. and G.K.; methodology, N.V.; software, N.V.; validation, G.K.; formal analysis, N.V.; investigation, N.V.; resources, G.K.; data curation, G.K.; writing—original draft preparation, N.V. and G.K.; writing—review and editing, N.V.; visualization, G.K.; All authors have read and agreed to the published version of this manuscript.

**Funding:** This research received no external funding.

**Data Availability Statement:** The data used in this study is freely available at https://github.com/NataliaVanetik/vacancy-resume-matching-dataset.

**Acknowledgments:** We wish to express our deepest gratitude to Yaarit Katzav, a senior HR manager at SCE Academic College, for taking the time to explain to us and our annotators how to match vacancies to resumes. Her guidance and valuable insights have been extremely helpful.

**Conflicts of Interest:** The authors declare no conflict of interest.

## Abbreviations

The following abbreviations are used in this manuscript:

| | |
|---|---|
| BERT | Bidirectional Encoder Representations from Transformers |
| BOW | Bag-of-Words |
| CV | Curriculum Vitae |
| HR | Human Resources |
| IR | Information Retrieval |
| IT | Information Technology |
| NE | Named Entity or Entities |
| NER | Named Entity Recognition |
| NLP | Natural Language Processing |
| RQ | Research Question |
| TF-IDF | Term Frequency-Inverse Document Frequency |
| VM | Vector Matching |

## Appendix A

*Appendix A.1. Named Entity Recognition Details*

The main types of NE that can be identified with spaCy appear in Table A1. Examples of named entities for both types of data appear in Table A2, together with their types (note that all NEs are lower-cased).

**Table A1.** NEs in spaCy.

| NE | Description |
|---|---|
| PERSON | Individuals, including names of people, fictional characters, or groups of people. |
| ORG | Organizations, institutions, companies, or agencies. |
| GPE | Geo-political entity that includes countries, cities, states, provinces, etc. |
| LOC | Non-GPE locations, such as mountains, bodies of water, or specific landmarks. |
| PRODUCT | Named products, including goods or commercial items. |
| EVENT | Named events, such as sports events, festivals, or conferences. |
| WORK_OF_ART | Artistic works, including books, paintings, songs, or movies. |
| LAW | Named laws, legislations, or regulations. |
| LANGUAGE | Named languages or language-related terms. |
| DATE | Dates or periods of time. |
| TIME | Specific times or time intervals. |
| PERCENT | Percentages or numerical ratios. |
| MONEY | Monetary values, including currencies, amounts, or symbols. |
| QUANTITY | Measurements or quantities. |
| ORDINAL | Ordinal numbers. |
| CARDINAL | Cardinal numbers or numerical values. |

**Table A2.** Named entity examples.

| Data Type | Named Entity | Type |
|---|---|---|
| Resume | Arlington | GPE |
| | mdm | ORG |
| | Over | CARDINAL |
| | 500 | CARDINAL |
| | The | DATE |
| | First | DATE |
| | Thirty | DATE |
| | Days | DATE |
| | The | DATE |
| | First | DATE |
| Vacancy | 1990 | DATE |
| | Israel | GPE |
| | Years | DATE |
| | Hebrew | LANGUAGE |
| | Masa | ORG |
| | English | NORP |

*Appendix A.2. Keyword Analysis for Developer and Non-Developer Vacancies*

Table A3 provides the lists of the top 10 keywords for the two types of vacancies used in automated experiments of Section 6.5. As can be seen from the table, the keyword phrases are different, and there are only four common keyword phrases.

**Table A3.** Top ten keywords for developer and data science vacancies.

| Developer Vacancies | |
|---|---|
| Keywords | Count |
| Software developer | 25 |
| Software development | 6 |
| Seeking software developer | 6 |
| Software developers | 5 |
| Senior software developer | 4 |
| Participate in scrum | 4 |
| Developer in Windsor | 4 |
| Development environment candidates | 4 |
| Agile development environment | 4 |
| Lead software developer | 3 |
| **Data Science Vacancies** | |
| Keywords | Count |
| Data scientist | 35 |
| Data scientist with | 7 |
| Data scientists | 6 |
| The data scientist | 5 |
| Seeking data scientist | 4 |
| Data analyst | 3 |
| Data scientist for | 3 |
| Senior data scientist | 3 |
| Data scientist role | 3 |
| Data engineers | 3 |

**Table A3.** *Cont.*

| | Data Science Vacancies | |
| --- | --- | --- |
| Keywords | | Count |
| Responsibilities kforce | | 1 |
| Responsibilities kforce has | | 1 |
| Software development engineer | | 1 |
| Talented software development | | 1 |

*Appendix A.3. The Effect of Different Sentence Embeddings*

We have compared different pre-trained sentence embedding models on resumes with human rankings and report the results in Table A4. In addition to `bert-base-uncased` [52], we tested the `nli-bert-large-max-pooling` [73], `longformer` [74], and `all-distilroberta-v1` [75] models. The top five results for every sentence embedding model are reported.

While none of these setups outperforms our best model, we can see that the scores improve significantly for `nli-bert-large-max-pooling` and `longformer` compared to our base model, `bert-base-uncased`. Therefore, we see the obvious benefits of using larger models when text representation is set to be sentence embedding.

**Table A4.** Comparing different sentence embedding representations.

| Resume Text Data | Vacancy Text Data | Sentence Embedding Model | Krippendorff's Alpha | Spearman's Correlation |
| --- | --- | --- | --- | --- |
| kw + summary | Summary | bert-base-uncased | 0.3695 | 0.3003 |
| ne + summary | Full text | bert-base-uncased | 0.3466 | 0.2773 |
| ne + summary | Summary | bert-base-uncased | 0.3155 | 0.2403 |
| Full text | Full text | bert-base-uncased | 0.3097 | 0.2340 |
| Full text | Summary | bert-base-uncased | 0.2917 | 0.2140 |
| Keywords + summary | Summary | nli-bert-large-max-pooling | 0.6211 | 0.5816 |
| Keywords + summary | ner + summary | nli-bert-large-max-pooling | 0.6151 | 0.5749 |
| ner + summary | Keywords + summary | nli-bert-large-max-pooling | 0.6139 | 0.5779 |
| Full text | Summary | nli-bert-large-max-pooling | 0.6070 | 0.5680 |
| Full text | Keywords + summary | nli-bert-large-max-pooling | 0.6040 | 0.5646 |
| ner + summary | Summary | Longformer | 0.5872 | 0.5467 |
| Keywords + summary | Summary | Longformer | 0.5806 | 0.5406 |
| Full text | Full text | Longformer | 0.5564 | 0.5138 |
| ner + summary | Keywords + summary | Longformer | 0.5517 | 0.5007 |
| Full text | Summary | Longformer | 0.5362 | 0.4900 |
| Full text | Keywords + summary | all-distilroberta-v1 | 0.5403 | 0.4867 |
| Keywords + summary | Summary | all-distilroberta-v1 | 0.5282 | 0.4808 |
| Keywords + summary | Keywords + summary | all-distilroberta-v1 | 0.4922 | 0.4408 |
| ner + summary | Keywords + summary | all-distilroberta-v1 | 0.4787 | 0.4198 |
| Full text | Summary | all-distilroberta-v1 | 0.4245 | 0.3672 |

*Appendix A.4. Extended Automatic Evaluation*

To further extend our automatically annotated evaluation of Section 6.5, we extracted 100 random developer vacancies from our dataset and 100 data scientist/senior vacancies and constructed 100 pairs of vacancies. For all these pairs, the ground truth has the form described in (1). We have evaluated our baselines and methods for all of the resumes against every one of the 100 vacancy pairs. The results are reported in Table A5; the top 20 results for the VM method variations are reported. We can see that all the reported VM methods demonstrate better performance than any of our baselines in all metrics, including accuracy; in fact, 19 of them achieve a perfect accuracy score of 1.0.

*Appendix A.5. Rankings Produced by ChatGPT API*

ChatGPT [76] is a language model developed by OpenAI [77], specifically referred to as a large language model (LLM). Trained using machine learning techniques, it can generate human-like text based on the input it is given. It analyzes the structure and content of the text it has been trained on, learns patterns, and then applies this knowledge to create new, contextually relevant content.

We used OpenAI API at https://platform.openai.com/docs/api-reference (accessed on 1 August 2023) to supply the LLM used by ChatGPT with five vacancies to evaluate and posed a query asking to rank these vacancies from the most relevant to the least relevant for every one of the 30 resumes we have human annotations for. The results of the comparison to human rankings appear in Table A6. We can see that although there is a positive correlation with human judgment in all three metrics, these scores fall far below the scores of the OKAPI BM25 algorithm and our top models in Table 8.

**Table A5.** Extended automatic tests.

| Baseline | Krippendorff's Alpha | Spearman's Correlation | Acc | | |
|---|---|---|---|---|---|
| OKAPI-BM25 | −0.0455 | −0.3939 | 0.3030 | | |
| BERT-rank | −0.2045 | −0.6061 | 0.1970 | | |
| **VM** | | | | | |
| Resume text data | Vacancy text data | Text representation | Krippendorff's | Spearman's | acc |
| Keywords + summary | ner + summary | tfidf | 1 | 1 | 1 |
| Keywords + summary | Job + description | tfidf + char ng + word ng + sbert | 1 | 1 | 1 |
| Keywords + summary | Job + description | tfidf | 1 | 1 | 1 |
| Keywords + summary | Job + description | Char ng | 1 | 1 | 1 |
| Keywords + summary | Job + description | Word ng | 1 | 1 | 1 |
| ner + summary | ner + summary | tfidf | 1 | 1 | 1 |
| ner + summary | Job + description | tfidf + char ng + word ng + sbert | 1 | 1 | 1 |
| ner + summary | Job + description | tfidf | 1 | 1 | 1 |
| ner + summary | Job + description | Char ng | 1 | 1 | 1 |
| ner + summary | Job + description | Word ng | 1 | 1 | 1 |
| Summary | ner + summary | tfidf | 1 | 1 | 1 |
| Summary | Job + description | tfidf + char ng + word ng + sbert | 1 | 1 | 1 |
| Summary | Job + description | tfidf | 1 | 1 | 1 |
| Summary | Job + description | Char ng | 1 | 1 | 1 |
| Summary | Job + description | Word ng | 1 | 1 | 1 |
| Doc | ner + summary | tfidf | 1 | 1 | 1 |
| Doc | Job + description | tfidf | 1 | 1 | 1 |
| Doc | Job + description | Word ng | 1 | 1 | 1 |
| ner + summary | Summary | tfidf | 0.9773 | 0.9697 | 0.9848 |

**Table A6.** Extended automatic tests on 30 human-ranked resumes.

| Baseline | Krippendorff's Alpha | Spearman Correlation |
|---|---|---|
| ChatGPT | 0.1422 | 0.0430 |

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
