# Peer review of "Job Vacancy Ranking with Sentence Embeddings, Keywords, and Named Entities"

_information, doi:10.3390/info14080468_

Round 1

Reviewer 1 Report

Job-vacancy ranking with sentence embedding, keywords, and name entities

1. In the first part of the introduction, the description of the challenges involved in lines 32-33 is not clear enough.

2. In the first part of the introduction, only a brief citation of the current state of the art is explained, without pointing out what specific shortcomings of previous methods, what problems are solved or how much the effect is improved by the method proposed in this paper in response to the shortcomings of others, and the lack of a summary of existing methods and the advantages of this method are explained.

3. In 2.1, the first step is to compute the text types, and the description here seems to be not clear enough, and I am confused about what result is obtained from the calculation here.

4. According to the meaning of 2.1 outline, your 2.2 summary should be intended to introduce four text types, your 2.2.1 and 2.2.2 introduce the full texts and the extractive summarization respectively, but your 2.2.3 and 2.2.4 introduce keyword extraction and named entity extraction, I am confused, where is the summaries enhanced by keywords or named entities you referred to in the previous article? Where is the enhancement reflected?

5. Has the method in this article been tried to experiment on other types of resume datasets? Because the title of the article does not limit the scope of occupation of the resume and job matching method, but the actual experiment is only conducted for computer development resumes.

6. In the summary of 4.3, I feel that the language logic is a bit confusing, and the labeling method used can be written more clearly here, and the method to evaluate the efficiency of the system should be explained to the readers.

7. From Section 6, it can be seen that the method in this paper does perform better than the existing resume vacancy matching algorithm, can you add a theoretical explanation of why such an effect occurs?

8. Figure 1 in the first paragraph of the introduction is supposed to be the overall flow chart, which is also mentioned in 2.1. Would it be more appropriate to put it in the method introduction section? We suggest to check the layout of the article again. Figure 2 is placed in subsection 2.2.1, is this figure only applicable to the text type of this subsection? The position of Table 1 in subsection 2.2.3 and Tables 2 and 3 in subsection 2.2.4 affects the reading experience very much.

None

Reviewer 2 Report

The article discusses a pipeline for finding a vacancy based on an unstructured resume. Experiments are carried out on resumes and vacancies of software engineers. Various variants of vector data representation models and various methods for extracting additional information from text are considered.

The proposed approach is described in detail and clearly, and in general is quite reasonable, albeit somewhat straightforward. The article is devoted to an interesting applied problem and makes a good impression. It can certainly be accepted, but only after a number of modifications. Below are a number of issues, shortcomings, questions and difficulties that were noted when reading the article.

1. Task formulation. It seems, that the most typical resume is a structural one, with the subsections, keywords, etc. However, authors consider only fully unsctructural case. Why it was chosen? How well can it be transferred to the sctructured data?

2. Vacancy dataset construction and usage. The most problematic part here is related with the "data science" vacancies. Firstly, both examples from tables 6 and 7 are clearly about software development which is very confusing. Secondly, the goal of this subset collection is not clear. It looks like the negative examples for the classification, however no real classification is provided on the full subset. The experiment with the attached examples is more about sanity check because "developer" and "data scientist" vacancies can be clealrly separated by the keywords (which were used to collect the dataset, by the way)

3. Approach. Confusing part is a simple concatenation of different text representations. It can be OK if this vector is then given to the network but it's not the case here. For instance, there is no reason to concatenate character n-gram with BERT since BERT already uses WordPiece representation. More discussion and explanation is needed here. Another suggestion is to try other encoders, such as RoBERTA, DeBERTA, BERT-Large, etc.

4. Baselines. Existing approaches are quite reasonable but they look very similar to the proposed approach. BM-25 is actually a slight modification of TF-IDF, and BERT ranking utilises the same model as VM. So, baseline a priori should lose the fight because it does not use keywords, NE and summary. More sophisticated approaches should come into play like LLM that are fine-tuned (or "prompted") on the similar tasks (IR, QA, etc.).

5. Evaluation. Of course, collecting human ratings is very time consuming but the actual dataset is extremely small. Possible recommendation here is to choose 3-4 best models, run them on the full dataset and then manually estimate relevance of the first 5-10 results (e.g., with the 3-grade scale) without making fully ordered list. Then repeat the same procedure for the baselines.

Round 2

Reviewer 1 Report

According to revisions provided by authors, the Manuscript is now acceptable to be published by Journal.

None.

Reviewer 2 Report

After reading the revised version of the manuscript I feel that the authors took into considerations all of my concerns. They extended descriptional part and performed additional experiments. The paper is ready to be published.